# Seismic Behavior of Concrete-Filled Circular Steel Tubular Column–Reinforced Concrete Beam Frames with Recycled Aggregate Concrete

**Zongping Chen** [1,2,*]**, Ji Zhou** [1]**, Zhibin Li** [1,3]**, Xinyue Wang** [1] **and Xingyu Zhou** [1]

1   College of Civil Engineering and Architecture, Guangxi University, Nanning 530004, China; 17687657937@163.com (J.Z.); bluesvictor@outlook.com (Z.L.); xinyuewangGXU@163.com (X.W.); zh-xy-59@139.com (X.Z)
2   Key Laboratory of Disaster Prevention and Structure Safety of Chinese Ministry of Education, Guangxi University, Nanning 530004, China
3   Faculty of Engineering, University of Auckland, 1023 Auckland, New Zealand
*   Correspondence: zpchen@gxu.edu.cn

**Abstract:** The application of recycled aggregate concrete (RAC) in concrete filled steel tubular (CFST) structures can eliminate the deterioration of concrete performance caused by the original defects of the recycled aggregate, which also provides an effective way for the recycling of waste concrete. In this paper, a test of a small scale model of a circular CFST column-reinforced concrete (RC) beam frame with RACs under low cyclic loading was presented in order to investigate its seismic behavior. The failure modes, plastic hinges sequence, hysteresis curve, skeleton curve, energy dissipation capacity, ductility and stiffness degeneration of the frame were presented and analyzed in detail. The test results show that the design method of the recycled aggregate concrete filled circular steel tube (RACFCST) frame complies with the seismic design requirements of a stronger joint followed by the stronger column and the weaker beam. The hysteresis curve of the frame is symmetrical, showing a relatively full shuttle shape; at the same time, the ductility coefficient of the frame is greater than 2.5, showing good deformation performance. In addition, when the frame is damaged, the displacement angle is greater than 1/38, and the equivalent damping ratios coefficient is 0.243, which indicates that the frame has excellent anti-collapse and energy dissipation abilities. In summary, the RACFCST frame has good seismic behavior, which can be applied to high-rise buildings in high-intensity seismic fortification areas.

**Keywords:** recycled aggregate concrete filled circular steel tube column; reinforced recycled aggregate concrete beam; frame; 100% replacement percentage; seismic behavior

## 1. Introduction

The use of recycled aggregate is one way to potentially extend the life of natural resources by supplementing their supply and reducing the environmental impacts of material extraction as well as the impact of construction demolition in landfills [1]. As a green building material, recycled aggregate concrete (RAC) can effectively deal with construction waste and realize the sustainable development of construction resources [2]. However, due to the accumulated internal damage caused by the secondary crushing of recycled coarse aggregate (RCA), the mechanical properties of RAC are weaker than those of natural aggregate concrete (NAC). Through a large number of experimental studies, scholars had confirmed that recycled concrete has a lower strength, elastic modulus, energy dissipation and durability but a larger peak strain, Poisson's ratio, shrinkage and creep, compared with NAC [3–11].

Therefore, improving the inherent defects of RAC to promote the research and application of RAC structure has become a hot issue in the field of concrete research.

In recent years, research on the combination of RAC and steel has been carried out gradually. Through the constraint effect of steel on RAC, the strength and deformation performance of RAC materials are improved, so as to realize the optimal combination of the two materials, and at the same time, construction waste is turned into treasure. Based on the mature research into concrete filled steel tube (CFST) structure, relevant scholars [12] have put forward the concept of a recycled aggregate concrete filled steel tube (RACFST) structure, that is, a new composite structure formed by RAC placed inside a steel tube. On the one hand, the core RAC is in the state of three-dimensional compression so the defects in the RAC's mechanical properties will be improved. On the other hand, the RAC can provide effective support to increase the stability of the external steel tube. Based on this theory, many scholars [13–21] have carried out experimental research on the mechanical properties of RACFST under different forces (e.g., axial compression, eccentric compression and cyclic loading). The results show that the bearing capacity of a RACFST column is similar to that of an ordinary CFST column, and that the static and seismic performance indices of a RACFST column (including a square steel tube and circular steel tube) were slightly different from those of an ordinary CFST column. Therefore, it is feasible to use RAC with reasonable preparation in a CFST, and a RACFST is an effective way to recycle waste concrete.

With the development of research and the urgent needs of engineering applications, engineers pay more attention to the seismic behavior of the RACFST frame in terms of whether it can meet the requirements of existing seismic design codes, which is an urgent question to be answered through research. However, at present, research on the RAC filled circular steel tube (RACFCST) frame structure is still scarce. Therefore, it is necessary to carry out the corresponding research on the RACFCST frame to reveal its real internal mechanisms and seismic behavior index. This paper firstly presents a test of a small scale model of a circular CFST column-RC beam frame with RACs under low cyclic loading in order to investigate its seismic behavior. Furthermore, the seismic behavior indices (e.g., load–displacement curves, failure behavior, stiffness degeneration, energy dissipation capacity and ductility) of the RACFCST frame were analyzed quantitatively to enrich the research regarding the RACFCST structure and provide a reference for the research, design and application of the RACFCST structure. Finally, the general behavior of the RACFCST frame was compared with that of the reinforced concrete (RC) frame proposed by Zhou et al [22].

## 2. Experimental Program

### 2.1. Specimen Details

In this test, one small-scale frame specimen was made with an RCA content equal to 100%, and the geometric ratio between the frame and the engineering prototype was 1:3. According to Chinese Standard GB50011-2010 [23], the required seismic design, including the construction details, was accomplished. The volume reinforcement ratio of the longitudinal reinforcement was 2.50% in the reinforced recycled aggregate concrete (RRAC) beam, and that of the stirrup in the encrypted area and non-encrypted area was 0.57% and 0.38%, respectively. No.10 I-steel with a flange width of 68 mm and web length of 100 mm was welded on the surfaces of both sides of the steel tube, with a length of 300 mm; a cover plate with a thickness of 8 mm was welded on the bottom of the steel tube at the end closest to the cast RAC; a cover plate with a thickness of 8 mm was welded at the top of the steel tube; and a RACFST column with a height of 100 mm was reserved for the effective transmission of axial force. The detailed dimensions of the frame are illustrated in Figure 1; it should be highlighted that most of the vertical force of the specimen is borne by the RAC of the core at the end of loading. In order to reflect the ultimate force transmission path and stress state of the specimen, the axial load ratio (*n*) selected in this paper is only related to the concrete, and *n* is equal to $N/(f_c A)$ and taken as 0.8, where *N* is the axial load, $f_c$ is the measured prism compressive strength of RAC with the Chinese standard

size of $150 \times 150 \times 300$ mm according to GB50010-2010 [24], and *A* is the cross-sectional area of column. In addition, the beam–column nodes of the RACFST frame adopt the form of rigid nodes connected by an external stiffening ring, as shown in Figure 2. It can be seen from the figure that the longitudinal bar in the beam was welded along the upper and lower flanges of the extended steel bracket to the junction of the reinforcing ring and the surface of steel tube, and the longitudinal bar and the upper and lower flanges adopted the form of a double-sided weld.

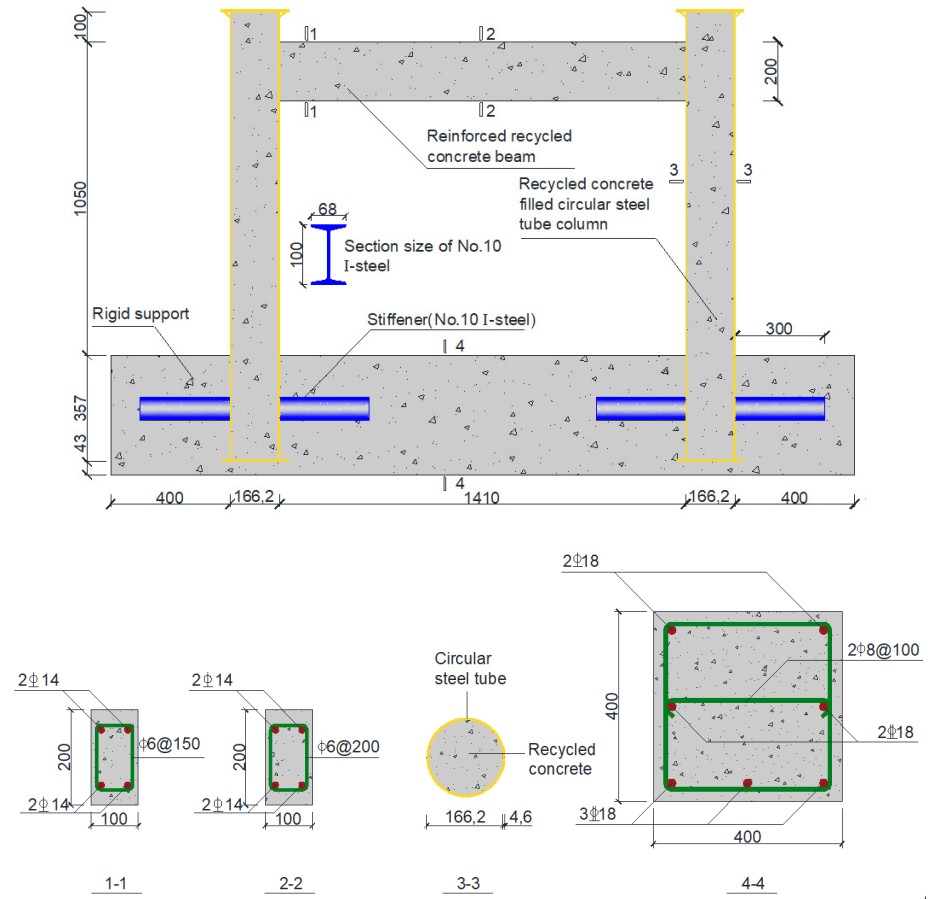

**Figure 1.** Specimen configuration and reinforcements.

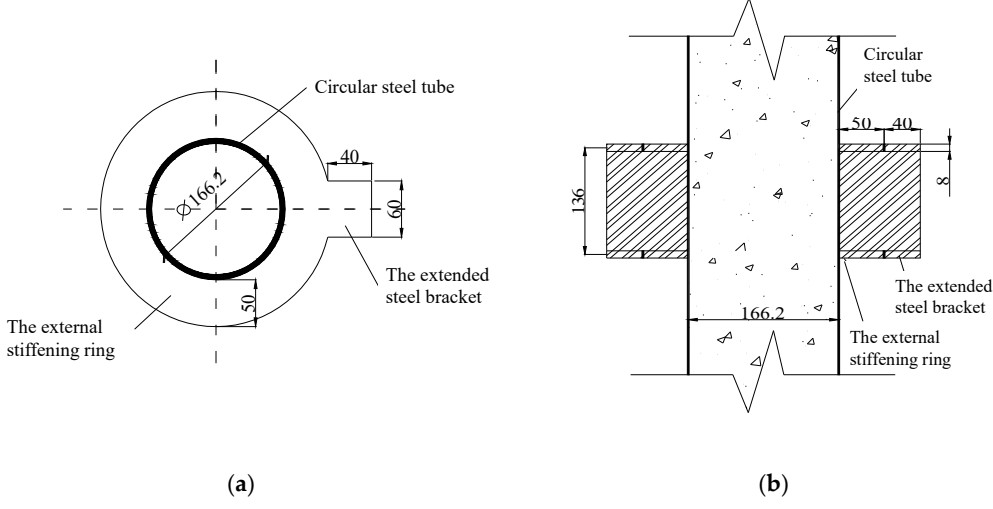

**Figure 2.** *Cont.*

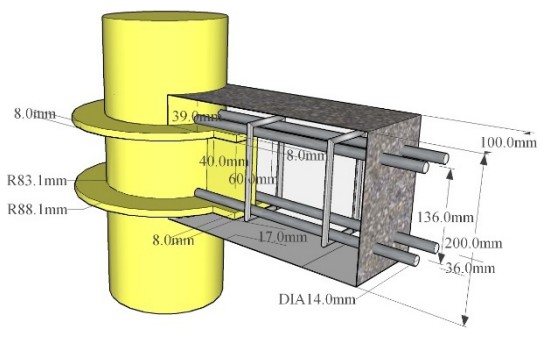

(**c**)

**Figure 2.** Details of the construction of the beam–column connection in (**a**) plane view, (**b**) elevation view and (**c**) spatial view.

### 2.2. Material Properties

The concrete mix proportions are given in Table 1. P.O 42.5 ordinary Portland cement, well-graded small and medium-size river sand and urban tap water were used in the test. The applied coarse aggregates were RCAs obtained from waste concrete specimens in the Key Laboratory of Disaster Prevention and Structural Safety of China Ministry of Education. RAC with a 100% replacement percentage of RCAs was used in the steel tube and beam, and the target concrete strength was set at about 50 MPa. Furthermore, the base of the frame was made of NAC. The physical properties of RCA and NCA are given in Table 2. By comparing the basic property indices of NCA and RCA, it is found that the physical property indices of RCA are quite different from those of NCA, which is mainly related to the composition of the material and the internal microstructure. A large amount of hardened cement mortar is attached to the surface of RCA, which makes its surface rough, with great porosity. In the process of the mechanical crushing of RCA, many closed microcracks or cracks are created inside, while the composition of NCA is homogenous, and the damage accumulated inside is less. Two steps were adopted to manufacture the beam-to-column frame; the first step was to infill the concrete into steel tubes and the second step was to place the concrete for the frame beam and basement beam. Hence, preparing the RAC in the frame was divided into two batches due to the sequence of fabricating the members of columns and beam. The measured mechanical properties of RAC in the steel tube and beam are listed in Table 3. The grade of the circular steel tube was Q235 and the outside diameter of tube and the wall thickness of steel plate were 166.2 and 4.6 mm, respectively; as for steel bars, HPB235 (a plain bar with a diameter of 6 mm) and HRB335 (a crescent ribbed bar with a diameter of 14 mm) were adopted as the stirrups and longitudinal reinforcement in the beam, respectively. The mechanical property indices were tested according to the Chinese standard GB /T228.1-2010 [25] and the measured value is shown in Table 4.

**Table 1.** The mix proportions of the concrete.

| Target Concrete Strength | Material Content (kg/m³) | | | | | |
| --- | --- | --- | --- | --- | --- | --- |
| | W/C | Cement | Sand | RCA | NCA | Water |
| 50 MPa | 0.47 | 520 | 628 | 1117 | 0 | 155 |

**Table 2.** The physical properties of the coarse aggregate.

| Coarse Aggregate | Grading (mm) | Apparent Density (kg/m³) | Bulk Density (kg/m³) | Water Absorption (%) | Water Content (%) |
|---|---|---|---|---|---|
| Natural | 5~20 | 2722 | 1435 | 0.05 | 0.00 |
| Recycled | 5~20 | 2655 | 1270 | 3.16 | 1.82 |

**Table 3.** The measured mechanical properties of recycled aggregate concrete (RAC).

| Member | Compressive Strength of Cube (N/mm²) | Compressive Strength of Prism (N/mm²) | Modulus of Elasticity (N/mm²) |
|---|---|---|---|
| Column | 53.8 | 48.6 | $4.24 \times 10^4$ |
| Beam | 47.3 | 37.7 | $3.56 \times 10^4$ |

**Table 4.** The measured mechanical properties of steel.

| Member | Yield Strength (N/mm²) | Ultimate Strength (N/mm²) | Modulus of Elasticity (N/mm²) | Yield Strain (με) |
|---|---|---|---|---|
| Steel tube | 406.5 | 478.3 | $2.18 \times 10^5$ | 1865 |
| Longitudinal reinforcement | 470.5 | 672.8 | $1.99 \times 10^5$ | 2364 |
| Stirrup | 419.9 | 548.5 | $2.16 \times 10^5$ | 1944 |

## 2.3. Test and Measuring Devices

The schematic diagram of the test device is illustrated in Figure 3a, and the corresponding field layout of the test device is shown in Figure 3b. In Figure 3a, No.1, No.2 and No.3 represent the reaction wall, vertical reaction steel column and reaction steel beam respectively, in which the steel beam and steel column are connected by high-strength bolts. No.4 represents the electro-hydraulic servo actuator that can push and pull back and forth, which is connected to the specimen by high-strength bolts through a special loading end. In addition, No.5, No.6, No.7 and No.8 refer to the hydraulic jack, specimen, pressure beam made of steel and roller device, respectively. It should be noted that a roller device is placed between the jack and the reaction steel beam to facilitate the free horizontal movement of the jack with the specimen.

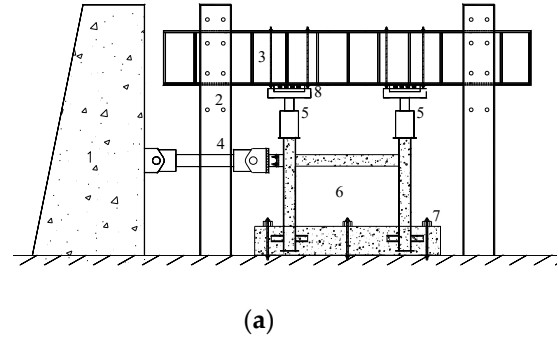
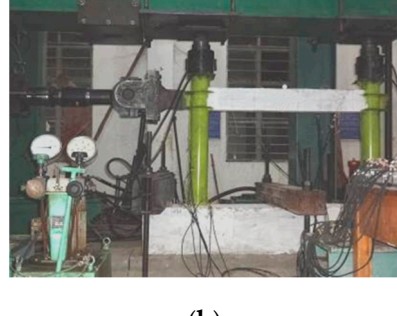

(**a**)             (**b**)

**Figure 3.** The test set-up arrangement of the (**a**) test device and (**b**) loading scene.

After the specimen was assembled according to the predetermined test axial pressure ratio, two identical hydraulic jacks were used to synchronously apply a constant vertical load to the top of the column, in order to simulate the actual situation in frame structures. After each of the vertical loads reached a stable value, the specimen was tested under a low-frequency cyclic lateral load. According to the Chinese standard of seismic test methods of buildings [26], the loading process included two main steps, namely a load-control step and a displacement-control step. In addition, in the load-control

step, each step was cycled once, and in the displacement-control step, each step was cycled three times, as depicted in Figure 4. The specific loading steps were as follows. First of all, the load-control was adopted for graded loading, with each 5 kN being one level, until the specimen reached the yield load $P_y$. After that, the displacement-control was used, taking the multiple of the yield displacement $\Delta_y$ as the grade difference to control the loading progress. The yield displacement was the horizontal displacement of the column end when the steel tube or longitudinal bar in the frame first reached the yield strain, which in this test was 6 mm. Finally, the test was finished when the load of the specimen dropped to about 85% of the peak load $P_u$. During the test, the horizontal load and displacement were measured by the load and displacement sensor of the electro-hydraulic servo loading system. The strain of steel tube, longitudinal reinforcement, stirrups and beam RAC was measured by strain gauges. The arrangement of the layout of the strain gauges on the steel tube and RAC beam is shown in Figure 5a, and the arrangement of the layout of the strain gauges on the longitudinal reinforcement and stirrups in the frame beam is shown in Figure 5b.

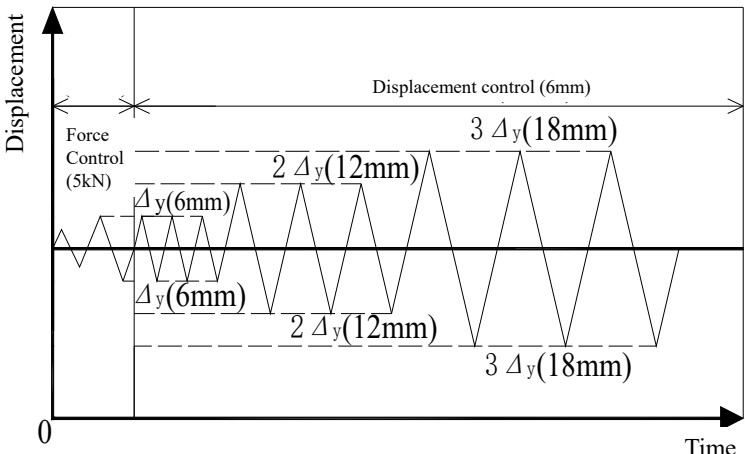

**Figure 4.** The loading system.

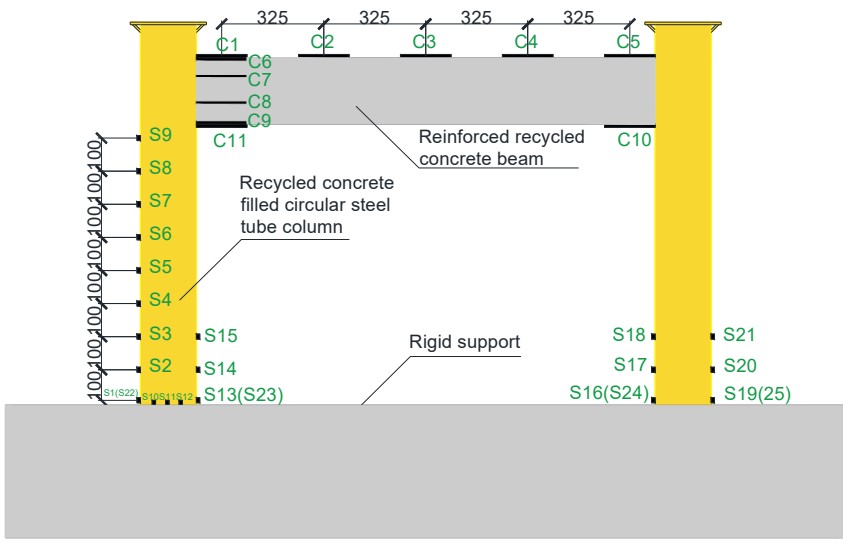

(**a**)

**Figure 5.** *Cont.*

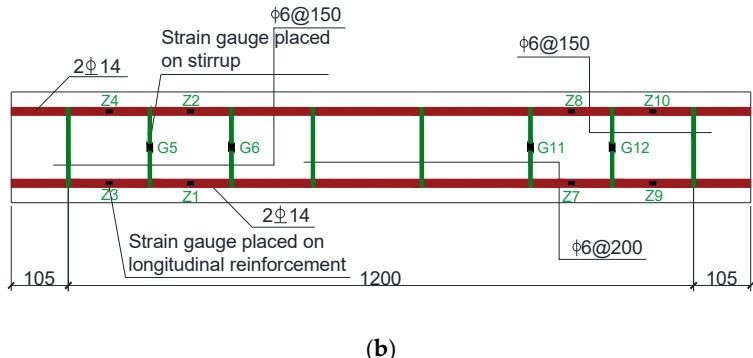

(**b**)

**Figure 5.** The arrangement of the strain gauges. (**a**) The layout of strain gauges on the steel tubes and RAC beam; (**b**) The layout of strain gauges on the longitudinal reinforcements and stirrups.

## 3. Test Process and Failure Characteristics

### 3.1. Test Process Description

For the convenience of description, the push and pull loadings in the test are defined as the positive (+) and negative (−) directions, respectively. At the same time, the side away from the loading point is the front side, the side close to the loading point is the rear side, the front side of the RRAC beam is to the right, and the back side is to the left. In the load-control stage, no cracks appeared on the surfaces of the beam until the lateral load reached ±30 kN. Then, bending cracks were found on the left and right sides at the end of the beam in the frame, and the length of the cracks was about 18 cm. When the specimen was loaded to ±40 kN, continuous bending cracks appeared on the right side and bottom of the beam, and the bottom cracks ran through the whole section. With the increase in load, the original cracks continuously extended upward and widened, and the new cracks gradually moved closer to the middle span of the beam. When the load was ±80 kN, the upper and lower cracks at the end of the beam passed through. Finally, small bending cracks appeared near the middle span of the beam, when the load was ±105 kN. Until the end of load-control stage, bending cracks were mainly distributed on the surface of the beam, with a gap spacing of 10~15 cm. It should be noted that the frame column was not bulging, but the measured strains of steel tube and the longitudinal reinforcements of beam were close to the yield strain.

In the displacement-control stage, when the displacement reached ±1$\Delta_y$, the bending cracks on surface of the beam continued to extend upward, which were generated at the load-control stage. At the same time, small oblique cracks began to appear at the beam end. When the displacement went up to ±2$\Delta_y$, the original bending cracks were no longer extended but widened, and the oblique crack continued to produce and develop, resulting in a cross oblique crack gradually forming on the surface of the beam. With an increase in the displacement up to ±3$\Delta_y$, the main cross diagonal cracks were formed at both ends of the beam, and the angle between them and the horizontal line was 33°~42°; this shows that the bending shear plastic hinge began to form gradually at this stage. At this time, the RAC at the beam end began to peel and fall off. With the increase in circulation time, the main oblique cracks continuously extended upward and widened, and new micro-cross oblique cracks were continuously generated. When the displacement reached ±4$\Delta_y$, the main cross diagonal crack had extended to the top and bottom of the beam. The RAC at the top and bottom of the beam began to show a horizontal crack under the influence of the diagonal crack, and the concrete protective layer began to break away. When the displacement reached ±5$\Delta_y$, large areas of RAC at the top and bottom of the beam were crushed and fell off in blocks. In some areas, the longitudinal bars and stirrups were exposed, and the deformation was serious. Therefore, it could be observed that an obvious bending shear plastic hinge was formed. By this point, the load had dropped to 85% of the peak load, and the specimen was severely deformed, so it was not feasible to continue loading, and the test was over. It should be noted that although the strain at the bottom of the steel tube had reached the yield strain

at the end of the test, the steel tube did not have bulges. The final failure mode of the RACFCST frame is shown in Figure 6.

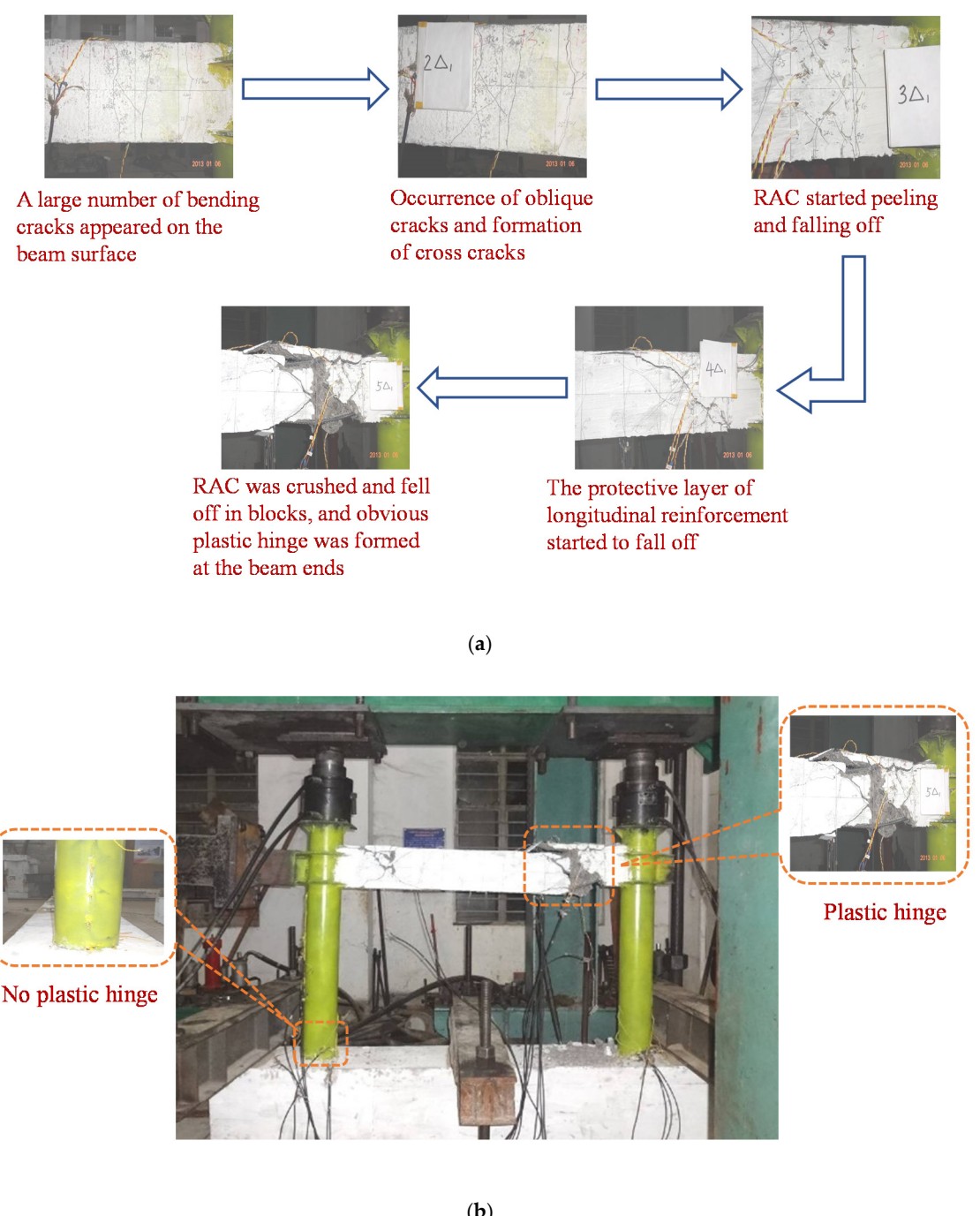

(a)

(b)

**Figure 6.** The failure modes. (**a**) The failure process of the beam; (**b**) The ultimate failure mode of the frame.

### 3.2. Analysis of Failure Characteristics

By analyzing the strain data obtained in the test and observing the failure mode of the specimen, the failure characteristics of the specimen were analyzed as follows:

(1) At the end of the beam of the RACFCST frame, bending cracks first appear, and then cross diagonal cracks evolve obliquely at 1/7~1/6 of the beam span and gradually develop into critical diagonal cracks. Finally, the RAC protective layer in the shear pressure area is lifted, forming a more

obvious bending shear plastic hinge. The strain at the bottom of the column reaches the yield strain, but the steel tube has no bulging and the plastic hinge is not obvious. In this test, the failure mode of beam results from bending shear failure, which meets the seismic design requirements of "strong shear weak bending". In the test of the RACFCST frame, the plastic hinge appears first in the beam and then in the column, which illustrates that the failure mechanism of the RACFCST frame structure results from beam hinge failure, which meets the seismic design requirements of "strong column and weak beam".

(2) If the yields of the bottom steel tube and the end longitudinal bar are taken as the yield marks of the RACFCST column and beam respectively, according to the measured data of the strain at the bottom of the steel tube and the end of the longitudinal bar, we know the sequence of the plastic hinges of the frame, and the sequence of the plastic hinges in the specimen is shown in Figure 7. It can be seen from this figure that under the action of push-pull, the sequence of plastic hinge generation is from the beam end to column bottom, which indicates that the full failure of the RRAC beam delays the appearance of the plastic hinge at the column bottom, and that this kind of new structure can meet the energy dissipation mechanism of the beam hinge.

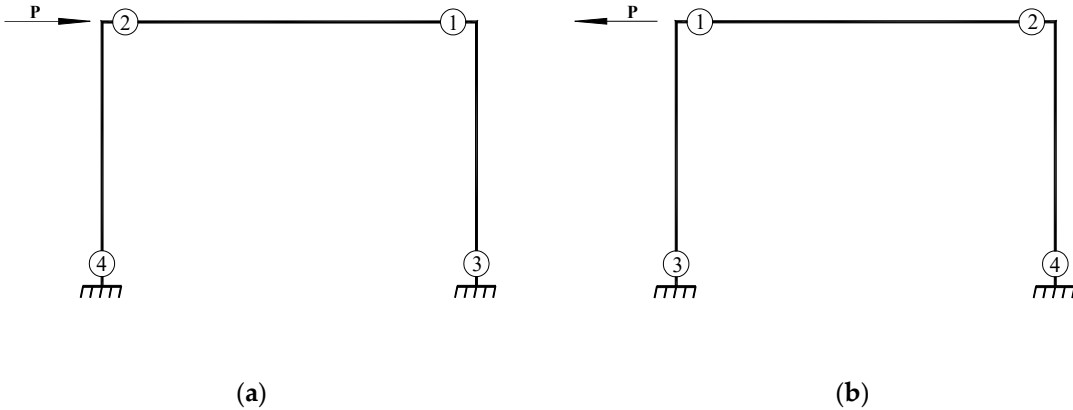

(**a**)  (**b**)

**Figure 7.** The sequence of the plastic hinges of the test frame under (**a**) positive loading and (**b**) negative loading.

(3) In the test, there is no vertical or diagonal crack at either end of the beam of the RACFCST frame, as shown in Figure 8. The reason is that the I-shaped extended corbel is arranged in this local area, and the corbel is welded with the stiffening ring outside the steel tube as a whole, which increases the strength and rigidity of the beam end and can effectively transfer the bending moment and shear force to the RACFCST frame column.

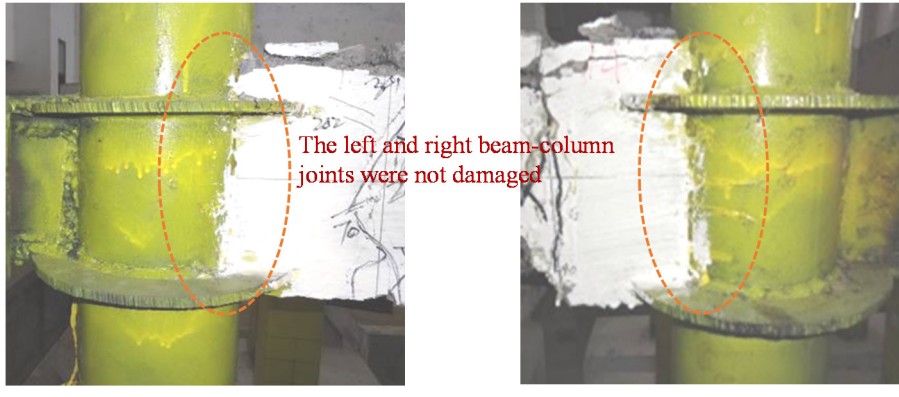

**Figure 8.** The core area of the joint.

(4) The RACFCST frame node area adopts the connection mode of the external stiffening ring. According to the test phenomenon, the core area of the node remains intact (see Figure 8). It complies with the seismic design requirements of "strong node, weak component".

## 4. Test Results and Analysis

### 4.1. Hysteresis Curve

The measured hysteresis curve of the frame is shown in Figure 9. It can be seen from the figure that the hysteresis curve of the RACFCST frame has the following characteristics:

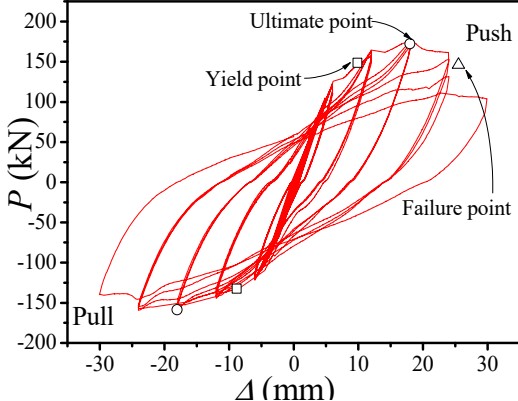

**Figure 9.** Hysteretic curves of the recycled aggregate concrete filled circular steel tube (RACFCST) frame.

(1) The hysteresis curve of the RACFCST frame is symmetrical and not pinched. The reason for this is that the lateral constraint of the external steel tube on the core RAC improves the strength and deformation performance of the core RAC, so that the cracking and crushing of the core RAC will not cause a sudden change in the lateral stiffness of the whole frame. Generally speaking, the hysteresis curve presents a relatively full spindle shape, and the energy dissipation capacity of the specimen is good.

(2) In the initial stage of load-control, the overall deformation of the frame is relatively small, the slope of the loading curve changes little, and there is basically no residual deformation after unloading. During the whole load-control stage, the hysteresis curve basically coincides, the hysteresis loop is not obvious, and the specimen is basically in the elastic stage.

(3) In the initial stage of displacement-control, there are many cracks at the end of the beam, and the steel bars in the beam yield first, which causes the hysteresis loop to fill gradually, and the residual deformation of the frame occurs after unloading. When the lateral displacement reaches about $\pm 2\Delta_y$, the frame begins to yield. After yielding, the hysteresis curve begins to incline to the displacement axis. With the increase in lateral displacement, the load acting on the specimen increases gradually, and the formed hysteresis loop becomes increasingly full. When the lateral displacement reaches $\pm 3\Delta_y$, the peak load of the RACFCST frame appears. At this stage, the three hysteresis loops deviate from each other more and the strength and stiffness of the specimen show obvious degradation. At the same time, the frame has significant residual deformation after unloading, which indicates that the specimen has acquired cumulative damage.

(4) When the lateral displacement goes up to $\pm 4\Delta_y$, the bearing capacity of the RACFCST frame decreases to the level of the failure load, but the lateral displacement of the specimen can still increase, which indicates that the energy dissipation capacity of the specimen has been fully developed. At this stage, the three hysteresis loops of the specimen have deviated greatly, the strength and stiffness degradation are very obvious, and the cumulative damage of the specimen has increased significantly.

### 4.2. Skeleton Curve

The skeleton curve of the specimen refers to the envelope formed by the peak points of the first cycle under each loading stage of the $P$-$\Delta$ hysteresis curve. It can clearly reflect the strength and deformation performance of the structure. The skeleton curve of the RACFCST frame is illustrated in Figure 10. It can be observed in the figure that the skeleton curve of the frame can be divided into three stages, namely the elastic stage, elastic-plastic stage and failure stage. In the elastic stage, the skeleton curve is approximately a straight line with a constant slope. After entering the displacement-control, a turning point appears in the curve, indicating that the specimen enters the elastic-plastic stage. At this point, the curve slope decreases continuously, which indicates that the overall stiffness of the frame decreases gradually. When the curve reaches the peak load, the plastic hinge at the beam end is very obvious, and the overall stiffness of the frame decreases to the minimum in the elastic-plastic stage. Then, the bearing capacity starts to decline, the displacement increases rapidly, and the frame structure enters the failure stage.

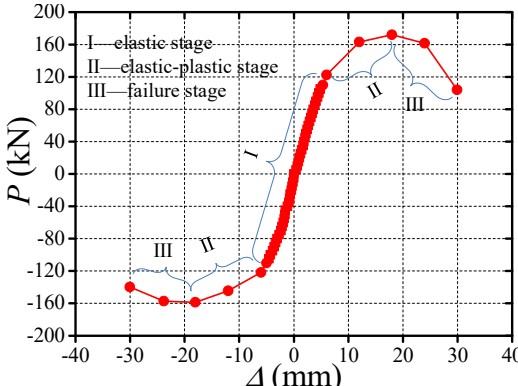

**Figure 10.** The skeleton curve of the RACFST frame.

### 4.3. Ductility

Ductility is an important performance index in seismic research on structural engineering. In this paper, the energy equivalent method is used to calculate the displacement ductility coefficient $\mu = \Delta_u/\Delta_y$, where $\Delta_y$ is the yield displacement and $\Delta_u$ is the limit displacement, which is taken as the corresponding displacement value when the load drops to $0.85P_u$. Figure 11 shows how the initial yield point of the specimen can be determined by the energy equivalence method. As shown in the figure, two lines $OY$-$YU$ are drawn to make the areas of the two shadows in the figure equal ($S_{OAB} = S_{BYU}$), and the $x$ coordinate of the intersection of the two lines $OY$-$YU$ and the skeleton curve is the initial yield displacement ($\Delta_y$). The calculated values of $\mu$ and each characteristic point are listed in Table 5, where $P_y$ and $P_f$ are respectively the load values corresponding to $\Delta_y$ and $\Delta_f$, and $\Delta_u$ is the displacement value corresponding to $P_u$. At the end of the loading, the positive load is reduced to $0.85P_u$, but the negative load is not, so only the positive ductility coefficient is given in the table, and the positive ductility coefficient of the frame is up to 2.58, which shows that the RACFST frame can meet the ductility requirements according to the design methods and construction measures.

**Table 5.** The measured load and displacement at characteristic points.

| Specimen Name | Loading Direction | Yield Point | | Ultimate Point | | Failure Point | | $\mu = \Delta_u/\Delta_y$ | $\mu_{average}$ |
|---|---|---|---|---|---|---|---|---|---|
| | | $P_y$ | $\Delta_y$ | $P_u$ | $\Delta_u$ | $P_f$ | $\Delta_f$ | | |
| RACFCST frame | + | 148.60 | 9.88 | 172.14 | 17.98 | 146.32 | 25.50 | 2.58 | 2.58 |
| | - | 132.48 | 8.83 | 158.62 | 18.02 | 134.83 | — | — | |

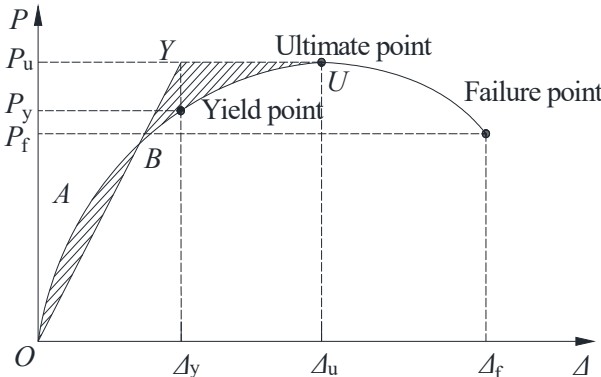

**Figure 11.** Characteristic points on the load–displacement curve.

Table 6 shows the inter-story displacement angles ($\theta_y$, $\theta_u$ and $\theta_f$) of the characteristic points of the skeleton curve. According to the relevant provisions of Chinese standard GB50011-2010 [24], "on the one hand, in the event of a moderate earthquake, the elastic inter story displacement angle of the structure shall be less than a specified limit value, 1/550 for reinforced concrete frame and 1/250 for multi-story and high-rise steel structure, so as to prevent the main structure of the building from being damaged and ensure the normal use function of the building". In this experiment, the displacement angle is 1/91 when the frame yields, which is far greater than the limit value of the standard, indicating that the deformation ability of the frame in the elastic stage can comply with the requirements well. On the other hand, in the event of a large earthquake, the elastic inter story displacement angle of the structure should be less than a specified limit value (1/50), so as to prevent the collapse of the structure. In this experiment, when the frame is damaged, the inter-story displacement angle is 1/33, which is far greater than the limit value of 1/50 specified in the standard, so the RACFST frame has a potent anti-collapse ability.

**Table 6.** Measured displacement angle at characteristic points.

| Specimen Name | Loading Direction | Yield Point | | Ultimate Point | | Failure Point | |
|---|---|---|---|---|---|---|---|
| | | $\theta_y$ | $(\theta_y)_{average}$ | $\theta_u$ | $(\theta_u)_{average}$ | $\theta_f$ | $(\theta_f)_{average}$ |
| RACFCST frame | + | 1/86 | 1/91 | 1/47 | 1/47 | 1/33 | 1/33 |
| | - | 1/96 | | 1/47 | | — | |

### 4.4. Energy Dissipation Capacity

The equivalent damping ratios—those are, $h_e = S_{(ABC+CDA)}/(2\pi \cdot S_{(OBE+ODF)})$, in which $S_{(ABC+CDA)}$ is the area of the hysteresis loop of DABCD, and $S_{(OBE+ODF)}$ is the total area of the triangles of OBE and ODF—for evaluating the energy dissipation capacity calculated from the load–displacement hysteretic loops are shown in Figure 12. The purpose was to reflect the better energy dissipation capacity of the frame proposed in this paper compared with that of the RC frame proposed by Zhou et al. [22]. Figure 13 gives the equivalent damping ratios versus the maximum lateral displacement corresponding to the test frame and the contrasting RC frame. It can be found that the value of $h_e$ increases with the increase in lateral displacement, which indicates that more energy is consumed. The reason is that the plastic hinge appears at the beam end and column end and develops continuously, which increases the energy dissipation capacity of the frame. When the lateral displacement is at the same level, the energy dissipation capacity of the RACFCST frame is better than that of RC frame. With the increase in displacement level, the gap in energy dissipation capacity between the two types of frame increases, which is mainly related to the good plastic deformation performance of the RACFCST columns in the later stage of the test. At the end of the test, the $h_e$ of the RC frame and RACFCST frame is 0.156 and

0.276, respectively, which shows that the energy dissipation capacity of the RACFCST frame is much better than that of the RC frame.

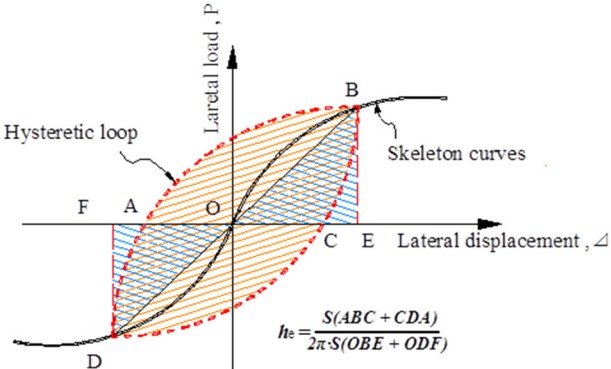

**Figure 12.** An energy dissipation capacity calculation diagram.

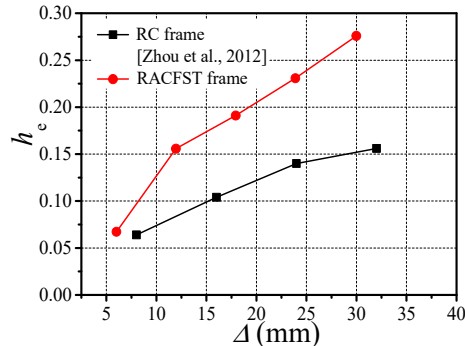

**Figure 13.** The energy dissipation capacity.

### 4.5. Stiffness Degradation

The secant stiffness $K_i$ is used to describe the stiffness degradation of the frame as shown in Figure 14, and the formula for calculating $K_i$ can be found in Chinese standard JGJ 101-96 [26] as

$$K_i = (|+P_i| + |-P_i|)/(| + \Delta_i|+| - \Delta_i|) \qquad (1)$$

where $P_i$ and $\Delta_i$ are the maximum values of the load and the corresponding displacement under the $i$th cycle, respectively.

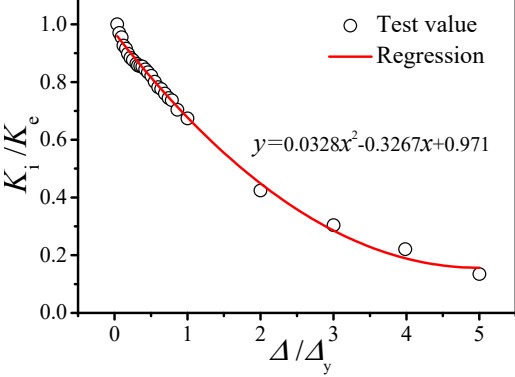

**Figure 14.** Stiffness degeneration.

It can be seen from Figure 14 that the secant stiffness decreases with the increase in displacement, and the stiffness degradation changes from fast to slow during the whole test. The main reason is that

before the yielding of the frame, there are many cracks on the beam, which makes the reduction rate of the bearing capacity less than the growth rate of the displacement, and the stiffness degradation is more rapid. However, when the plastic hinge appears at the beam end and column end, the good deformation capacity of the frame causes the bearing capacity to decrease slowly. At this point, the reduction rate of the bearing capacity is less than the growth rate of the displacement, and the stiffness degradation is relatively slow. At the same time, it can be seen from the figure that the stiffness degradation of the frame has obvious regularity. The RACFCST frame is distributed in the form of a quadratic function, and the mathematical expression is as follows:

$$y = ax^2 + bx + c \tag{2}$$

where $y$ is $K_i/K_e$; $x$ is $\Delta/\Delta_i$; $K_e$ is the elastic stiffness, which is calculated by the initial load divided the displacement corresponding to the initial load in the load-control stage; and $a$, $b$ and $c$ are the regression factors, which are equal to 0.0328, −0.3267 and 0.971, respectively.

## 5. Conclusions

This article discusses the experimental results of the seismic behavior of concrete-filled circular steel tubular column–reinforced concrete beam frames with recycled coarse aggregate. The results show that:

(1) For the RACFCST frame designed in this paper, the RRAC beam finally suffered from bending shear failure, and the plastic hinge appeared first in the beam and then in the column, which illustrates that the failure mechanism of the RACFCST frame structure results from beam hinge failure. In addition, the joints constrained by the external stiffening ring were not damaged from beginning to end. Therefore, the frame is characterized by the concept of "strongest joint, stronger column, and weaker beam".

(2) The hysteresis curve of the RACFCST frame is symmetrical and has good stability. During the test, the hysteresis curve is not pinched and presents a full shuttle shape. Furthermore, the $h_e$ of the RACFCST frame at the yield point and peak point is larger than that of the RC frame. All of the above illustrate that the RACFCST frame has a good energy dissipation capacity.

(3) The ductility coefficient of the RACFCST frame is more than 2.5, which indicates that the frame has good ductility, and that the ductility of the frame meets the requirements of the Chinese standard.

(4) When the RACFCST frame is damaged, the inter-story displacement angle is 1/33, which is far greater than the limit value of 1/50 of the Chinese standard regarding RC frames and multi-story and high-rise steel structures, so the RACFCST frame has a strong anti-collapse ability.

**Author Contributions:** Z.C. conceived the experiments, J.Z., Z.L. and X.W. wrote the initial draft of the manuscript. Z.C. and J.Z. analyzed the data and wrote the final manuscript, X.Z. contributed to the revision of this paper. All authors have read and agreed to the published version of the manuscript.

**Funding:** This research report was financially supported by the National Natural Science Foundation of China (No. 51578163), the Key Project of Natural Science Foundation of Guangxi Province (No. 2016GXNSFDA380032), the Key R & D project in Guangxi (2017AB02006) and Special fund project for "Bagui" scholars ([2019] No.79).

**Acknowledgments:** The authors are very grateful for the support of the above project funds.

**Conflicts of Interest:** The authors declare no conflict of interest.

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
