# Peer review of "Seismic Behavior of Concrete-Filled Circular Steel Tubular Column–Reinforced Concrete Beam Frames with Recycled Aggregate Concrete"

_applsci, doi:10.3390/app10072609_

Round 1
Reviewer 1 Report
This paper is a very interesting topic.
-----------------------------------------------------------------------------
Cyclic behavior of concrete-filled steel tubular column–reinforced concrete beam frames incorporating 100% recycled concrete aggregates, Advances in Structural Engineering, 21(10), 2018
------------------------------------------------------------------------------
I would like to ask you for an explanation of the originality and the differences from the above paper(another your paper etc).
The pictures inserted in the paper are the same.
If it is a duplicate, please retract it.
Author Response
Response to Reviewer 1 Comments
Point 1: This paper is a very interesting topic.
-----------------------------------------------------------------------------
Cyclic behavior of concrete-filled steel tubular column–reinforced concrete beam frames incorporating 100% recycled concrete aggregates, Advances in Structural Engineering, 21(10), 2018
------------------------------------------------------------------------------
I would like to ask you for an explanation of the originality and the differences from the above paper(another your paper etc).
The pictures inserted in the paper are the same.
If it is a duplicate, please retract it.
Response 1: I'm glad to discuss this issue with you. As the reviewer said, I published a similar research article in the Journal of “advanced in structural engineering” that is simply called article A. However, there are essential differences between article A and the paper submitted to the Journal of “Applied Science” that is simply called article B, which are mainly reflected in the following aspects:
i) In article A, the main research object is square concrete-filled steel tubular column–reinforced concrete beam frame realized employing 100% recycled coarse aggregates, while in article B, the main research object is concrete-filled circular steel tubular column–reinforced concrete beam frames with recycled aggregate concrete, so there are differences between square and circle in the section form of frame column. As we all know, there is a great difference in the mechanical properties between square CFST columns and circular CFST columns, which is one of the important reasons to study the seismic behaviour of concrete-filled steel tubular column–reinforced concrete beam frames incorporating 100% recycled concrete aggregates with different cross-sections. At the same time, it also shows that article A and article B are a series of results obtained through experimental research.
ii) The joint design and treatment of frame structure is always a problem worthy of attention. When it is connected with reinforced concrete beam, the design of joints of two different sections of CFST column is quite different, which makes the performance of frame structure at the joint also quite different. This is another main reason for this study. Comparing article A with article B, it can be found that there are great differences in the design of beam column joints.
iii) The difference of research object determines the actual content of the two papers is different. Of course, the seismic performance analysis of the frame in the paper is similar, which can not be avoided. Both of them analyze and discuss the seismic performance parameters of their research objects, but the contents are absolutely unique.
iiii) A small number of pictures in article A and B are repeated, but the author is only for simple comparative analysis of the seismic performance of CFST column frame with different cross-section forms, and does not analyze the seismic performance of the concrete-filled circular steel tubular column–reinforced concrete beam frames with recycled aggregate concrete in detail, which is a major lack of this experimental study. Therefore, the author wants to complete this part through article B, and provide reference for scholars in the industry.
iiiii) In order to avoid the same misunderstanding to the readers, the author redraws the pictures in article B to prevent the direct copy of the pictures. Here, the author apologizes for the misunderstanding caused to you in the earlier stage!

Reviewer 2 Report
Please see the file attached

Author Response
Response to Reviewer 2 Comments
Point 1: Line 54 Please, indicate the reference relative to the reinforced concrete (RC) frame reported in the existing literature.
Response 1: According to the reviewer's opinion, the reference information has been completed, and the reference number has been reordered.
Point 2: Figure 1: Section 4-4 is not indicated in the specimen configuration layout.
Response 2: The position of section 4-4 in the specimen configuration layout is supplemented.
Point 3: Line 64 “I-10 steel were welded…” It is not clear that I-10 steel are the steel profiles, specify it better.
Response 3: According to the reviewer's suggestion, the specific dimensions and cross-sectional shapes of I-10 steel are provided in the paper.
Point 4: Line 72 Please, report the Chinese Standard Normative reference for the determination of prism compressive strength.
Response 4: The author has supplemented the name of the applied specification in the corresponding position.
Point 5: Figure 9: Please, the unit of measurement shall be indicated in curly brackets.
Response 5: It has been revised according to the reviewer's comments.
Point 6: Line 246 Three stages, namely elastic stage, elastic-plastic stage and failure stage relative to the skeleton curve are detailed in the text. For clarity, it is suggested to indicate such stages in figure 10.
Response 6: The author has supplemented each stage in Figure 10.
Point 7: In the conclusion it is reported the sentence:
(3) The ductility coefficient of RACFCST frame is more than 3, which…
A similar sentence is specified in the Introduction. By the ductility analysis reported in section 4.3, the performance index calculated considering just positive loads is 2.58. Better specify this point because it is concluded that the ductility value is more than 3.
Response 7: This is a writing error and the author has corrected it. The specific modifications are as follows “The ductility coefficient of RACFCST frame is more than 2.5, which…”.
Point 8: Please, the following references must be added to improve the state-of-the-art on different aggregate/reinforcement for concrete treated in the paper:
“Mechanical behavior of concretes made with non-conventional organic origin calcareous aggregates”, Construction and Building Materials, Vol. 179, 2018, Pages 100-106.
Response 8: After carefully reviewing the references recommended by the reviewers, the author considers this research to be of reference significance and therefore cites it in this article.

Round 2
Reviewer 1 Report
Thank you for your faithful reply.